# Revision Total Knee Arthroplasty Utilizing Threaded Pins in Cement for Tibial Bone Loss

**DOI:** 10.3390/medicina59010162

**Published:** 2023-01-13

**Authors:** Max Jiganti, Nicholas Tedesco

**Affiliations:** Good Samaritan Regional Medical Center, Corvallis, OR 97330, USA

**Keywords:** revision total knee arthroplasty, metaphyseal cone, cemented stem, tibial bone loss, Steinmann pin

## Abstract

*Introduction*: a primary goal in revision total knee arthroplasty is to recreate and restore near-normal knee biomechanics by reapproximating the native anatomy. Tibial bone loss poses a challenge for surgeons. Bone cement, bone allograft, screws-in-cement, metaphyseal sleeves or cones, and metallic augments are some options for addressing bony deficiency, with endoprosthetic proximal tibia replacement a consideration for the most severe cases. *Case Description*: we present a case for the novel use of threaded Steinmann pins augmented with cement to reconstruct a massive tibial metaphyseal cortical defect during revision knee arthroplasty. A 76-year-old male presented with an infected primary total knee arthroplasty using MSIS (Musculoskeletal Infection Society) criteria and underwent a standard two-stage revision total knee arthroplasty once the knee was confirmed sterile. Intraoperatively, significant posteromedial and metaphyseal tibial bone loss was identified. In order to avoid proximal tibial replacement and the extensor mechanism complications seen with these, coupled with obligate gastrocnemius flap, a metaphyseal cone was utilized in the proximal tibia with four vertical threaded Steinmann pins spaced approximately 1 cm apart at its periphery, subsequently cut flush with the level of the cone after cementation to recreate the tibial cortex. The patient’s function and range of motion continue to improve with no evidence of structural complication at 2.5 years of follow-up. *Discussion*: the implementation of threaded Steinmann pins was utilized in this case to stabilize a cemented metaphyseal cone in the revision of an infected total knee with significant tibial bone loss. The threaded property may help prevent migration of these pins in comparison to smooth pins. Creation of a stable platform in a revision total knee arthroplasty poses a substantial challenge in the context of significant bone loss, and our case depicts a good short-term outcome and another option for surgeons to consider before moving toward endoprostheses.

## 1. Introduction

Attaining adequate alignment and stability in revision total knee arthroplasty is essential for providing a patient with a functional joint [1,2]. Bone loss poses a challenge in creating stability at the bone-implant interface in cases with large bone defects. Multiple surgical options exist to address these bone defects, such as bone cement, bone allograft, screws-in-cement, metaphyseal sleeves or cones, and metallic augments, with endoprosthetic proximal tibia replacement a consideration in the most severe cases. However, endoprostheses have increased complication rates relating to hardware failure, aseptic loosening, soft tissue coverage problems, and infection [3,4,5]. Modular metal augments such as metaphyseal sleeves fill bone defects and are superior to cement alone [6]. Cement may cause thermal necrosis and adding a metaphyseal cone limits the quantity of cement needed to fill the bone defect. The advantage of metaphyseal metal augments compared to an endoprosthetic is the avoidance of resecting further bone surrounding the defect. The use of cement surrounding K-wires within bone defects has shown early success [7] in the literature, and provides surgeons with an option before moving up the treatment algorithm to an endoprosthesis for proximal tibial bone loss. We discuss a case of revision total knee arthroplasty treated surgically with threaded pins within cement surrounding a metaphyseal cone to address significant proximal tibia bone loss.

## 2. Case Description

A 76-year-old male presented to our institution with an infected primary total knee arthroplasty (performed at an outside institution) using MSIS criteria, with positive cultures for group B streptococcus. He subsequently underwent an irrigation and debridement with retention of components by the on-call trauma surgeon. Three days later, our team assessed the patient’s knee X-rays (Figure 1) and took him to the operating room for the first of a planned two-stage revision (Figure 2). The components were noted to be press-fit, and these were removed with an oscillating saw and bone tamp. A large posterior wall defect on the medial tibia was visualized upon extraction. A Biomet stage I-articulating spacer was then impacted with excellent fit. Post operatively the patient was placed on partial weightbearing precautions and received 6 weeks of intravenous cefazolin.

Two months later, after confirmation of sterility via repeat Synovasure Alpha Defensin ELISA (Enzyme-linked Immunosorbent Assay) Test, the patient was brought back to the operating room with plan to address the known tibial bone loss with both cement and Steinmann pins. A medial parapatellar approach along with a V-Y advancement quadricepsplasty was chosen for the revision. The patient’s index total knee arthroplasty was performed at an outside institution. The previously visualized significant posteromedial and metaphyseal tibial bone loss was identified after the spacer was removed. To address the bony defect, a metaphyseal cone was inserted into the proximal tibia with good fit, though without clear axial stability. To enhance the strength of the construct to support weightbearing, four vertical threaded Steinmann pins were inserted into the metaphyseal bone approximately 1 cm apart at the periphery (Figure 3). The pins were cut at the superior level of the cone after cementation to recreate the tibial cortex, and the tibial tray was subsequently placed (Figure 4). The patient’s postoperative course was uneventful and there was no evidence of repeat infection or wound healing issues. Two-year follow up X-rays were obtained (Figure 5). The patient retains good ambulatory function and range of motion at his knee at 3 year-follow up, with no signs of instability or evidence of infection. At this time point, his global mental and physical health utilizing patient reported outcome measure scores (PROMIS) were 67.7 and 56 for PROMIS-Mental Health and PROMIS-Physical Health, respectively.

## 3. Discussion

Revision total knee arthroplasty for patients with significant tibial bone loss proves a challenge for surgeons to create a mechanically stable knee. The literature suggests a variety of techniques to create stability within the defect such as screws in cement, metal augments, impaction bone grafting, structural allografts and endoprostheses [3,8,9,10,11,12,13]. With larger bony defects, the difficulty increases to create a stable platform for weightbearing.

A major goal in revision total knee arthroplasty is to preserve as much native bone as possible, while constructing a durable joint that can withstand cyclical loading. In significant tibial bone loss, especially in tumor surgery, endoprostheses are commonly turned to in these scenarios. These larger prosthetics have significant risks that are well known in the literature to include higher complication rates [3,4,5]. A comparative statistical analysis from 2015 of endoprostheses reported a 36% failure rate of these tibial components within four years, with the top three causes being soft tissue failure, aseptic loosening, and structural failure [14]. A patient-specific approach must be used when considering these implants, and these higher rates of complications with endoprostheses implore surgeons to build an armamentarium of options to consider prior to escalating to larger endoprostheses when possible.

The use of K-wires or pins within cement to increase bony stability has been sparsely described in the literature. Bilgen et al. describe adequate results for smooth K-wires in the setting of large medial tibia bone defects (>20 mm) in primary total knee arthroplasty at two-year follow up [7]. We have attempted two cases of utilizing smooth pins to address bone loss in revision total knee arthroplasty at our institution, yet issues emerged with the backing-out of the pins. In the presented case, the threaded nature of the pins may have helped in avoiding this complication, as no pin malposition was observed during the postoperative follow up. Our decision to use four pins was based on the examination of the tibial defect and to evenly space the pins while maximizing cementation around both the cone and pins.

Further research such as biomechanical studies are needed to identify the ideal pin spread in cement to create the strongest construct. Additionally, higher level of evidence including prospective studies and randomized controlled trials are needed to fully evaluate the effectiveness of this technique. A longer follow up for this technique is also necessary in terms of counseling patients regarding expectations. Some benefits of utilizing threaded pins may be the relatively simple nature of the procedure due to smaller implants compared to interventions utilizing screws, larger bone grafts, or the technical nature of endoprostheses. Additionally, the implementation of threaded pins compared to screws or other larger implants can help to diminish the overall cost of the procedure without sacrificing the stability of the joint.

## 4. Conclusions

Addressing uncontained tibial defects in the setting of revision total knee arthroplasty poses a challenge for surgeons attempting to restore stability and joint mechanics. We present a successful short-term outcome for threaded pins in cement to address significant tibial bone loss in a patient undergoing a revision total knee arthroplasty. The benefits of this procedure include the relatively shallow learning curve when comparing alternative methods of treatment, in addition to the lower cost without hindering the stability of the construct.

## Figures and Tables

**Figure 1 medicina-59-00162-f001:**
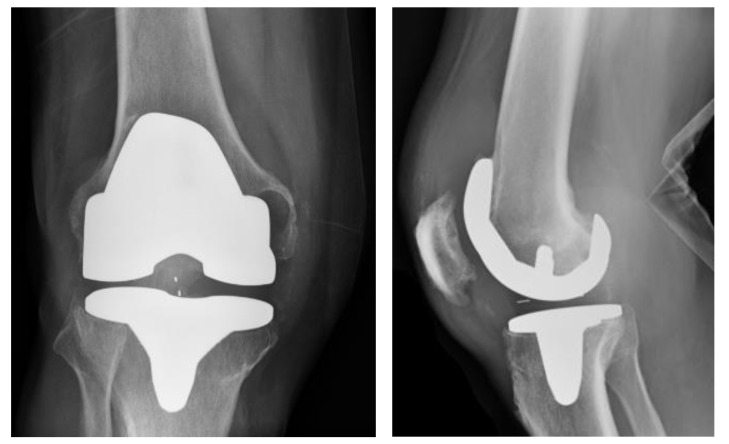
AP (Anteroposterior) and lateral radiographs of infected primary TKA (total knee arthroplasty).

**Figure 2 medicina-59-00162-f002:**
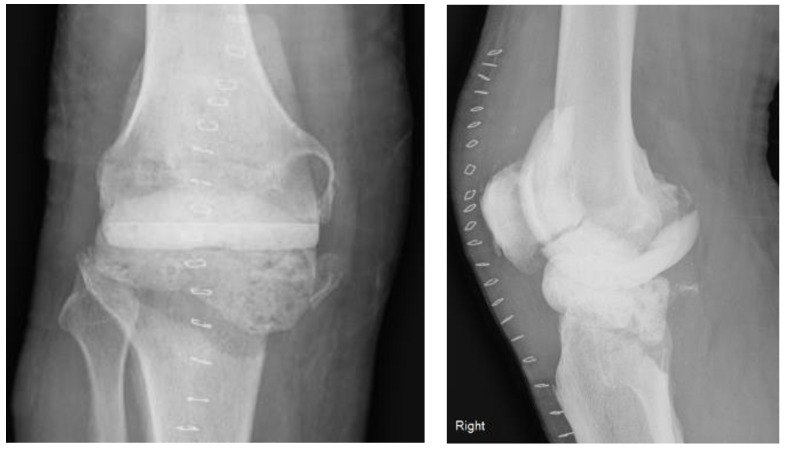
AP and lateral radiographs of antibiotic spacer.

**Figure 3 medicina-59-00162-f003:**
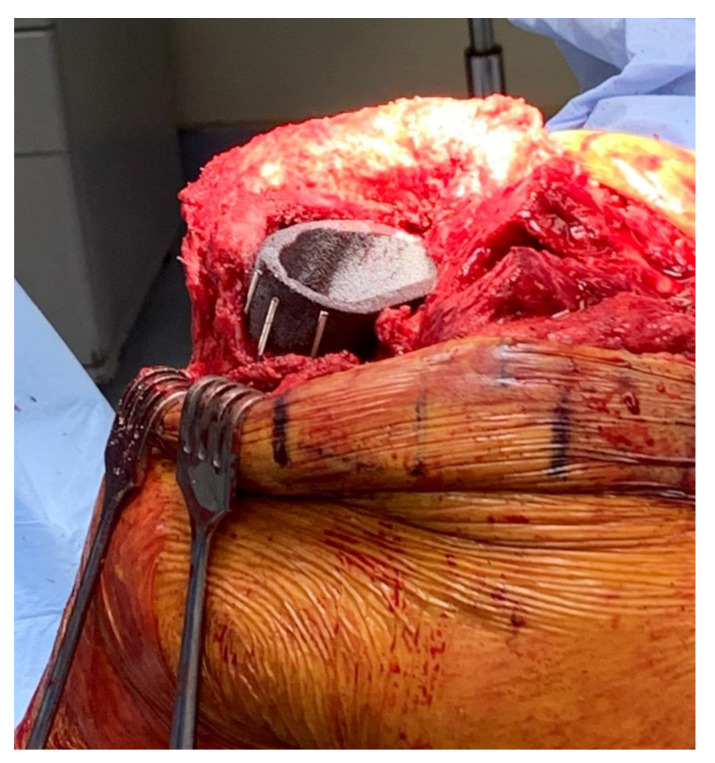
Depiction of four threaded Steinmann pins placed vertically into the proximal tibia, surrounding the metaphyseal cone.

**Figure 4 medicina-59-00162-f004:**
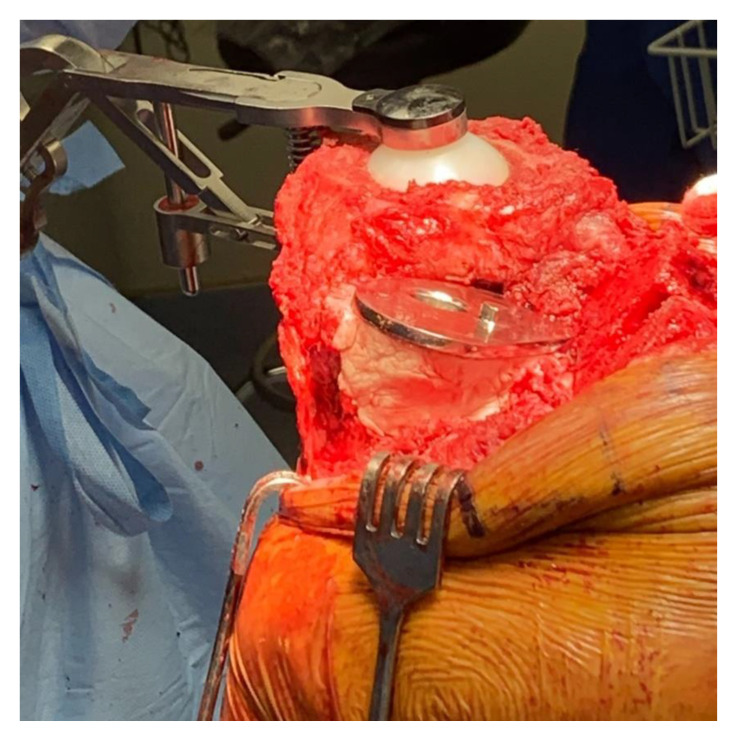
Underneath the tibial component, cement encompasses the metaphyseal cone and threaded Steinmann pins.

**Figure 5 medicina-59-00162-f005:**
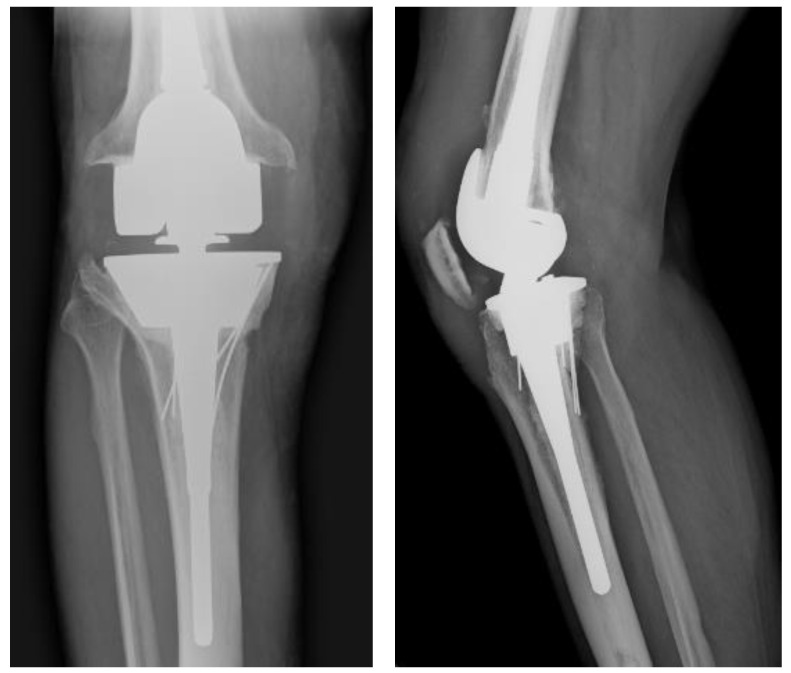
Two year post operative AP and lateral X-rays.

## Data Availability

No new data were created or analyzed in this study. Data sharing is not applicable to this article.

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
