# Peer review of "Revision Total Knee Arthroplasty Utilizing Threaded Pins in Cement for Tibial Bone Loss"

_medicina, 2023, doi:10.3390/medicina59010162_

Round 1

Reviewer 1 Report

GENERAL COMMENTS:

Dear Authors, I think this is an interesting work, which might be worth publishing after careful review. The future of replacement surgery will be the revision setting, which will increase in the next years. Accurately describing a potentially simple and inexpensive technique could be important.

SPECIFIC COMMENTS:

INTRODUCTION

- I recommend adding some references to the first part of the introduction about alignment:

  • DOI: 10.1016/j.otsr.2017.07.010
  • DOI: 10.1007/s00167-018-5017-0

CASE DESCRIPTION

- How was the patient studied preoperatively (X-Ray? CT?)? Was preoperative planning planned? 

- By Synovasure what do you mean? (Synovasure Alpha Defensin ELISA Test?)

- Did you use any functional scores to evaluate the patient in the follow-up? 

IMAGES

- I recommend cropping images to avoid seeing people or objects in the background

- In addition to the intraoperative images, it would be important to add pre-operative and post-operative radiographic images

DISCUSSION

- Are there specific tips and tricks related to this technique?

- Are there specific possible complications related to this technique?

GRAMMAR

- Line 10: Modify as suggested: Steinmann (not Steinmeann)

- Line 59-60: Modify as suggested: "A medial parapatellar approach along with a V-Y advancement quadricepsplasty was chosen for the revision" (not "The right knee was entered via a medial parapatellar approach along with a right knee V-Y advancement quadricepsplasty")

- Line 64: Modify as suggested: axial (not axially)

- Line 98: Modify as suggested: describe (not describes)

Reviewer 2 Report

1. Please adhere to CARE guidelines that provide comprehensive understanding on the accuracy, transparency, and usefulness of case reports. 

2. Please provide a filled-in CARE checklist encompassing all the necessary items in your case report.

3. The case report could be more strong if objective measures of function are evaluated such as scores, etc.

4. The current evaluation such as good ambulatory function and range of motion are subjective.

Round 2

Reviewer 1 Report

After the revisions, the manuscript is improved and the case description is more accurate. The addition of preoperative and postoperative radiographs and the postoperative clinical evaluation provide a greater understanding of the technique.